# Integrative Transcriptomic Analysis Reveals Distinctive Molecular Traits and Novel Subtypes of Collecting Duct Carcinoma

**DOI:** 10.3390/cancers13122903

**Published:** 2021-06-10

**Authors:** Chiara Gargiuli, Pierangela Sepe, Anna Tessari, Tyler Sheetz, Maurizio Colecchia, Filippo Guglielmo Maria de Braud, Giuseppe Procopio, Marialuisa Sensi, Elena Verzoni, Matteo Dugo

**Affiliations:** 1Platform of Integrated Biology, Department of Applied Research and Technology Development, Fondazione IRCCS Istituto Nazionale dei Tumori, 20133 Milan, Italy; chiara.gargiuli@istitutotumori.mi.it; 2Department of Medical Oncology, Fondazione IRCCS Istituto Nazionale dei Tumori, 20133 Milan, Italy; pierangela.sepe@istitutotumori.mi.it (P.S.); filippo.debraud@istitutotumori.mi.it (F.G.M.d.B.); giuseppe.procopio@istitutotumori.mi.it (G.P.); elena.verzoni@istitutotumori.mi.it (E.V.); 3Department of Cancer Biology and Genetics, College of Medicine and Comprehensive Cancer Center, The Ohio State University, Columbus, OH 43210, USA; anna.tessari@osumc.edu (A.T.); tyler.sheetz@osumc.edu (T.S.); 4Department of Urology, The Ohio State University Wexner Medical Center, Columbus, OH 43210, USA; 5Department of Pathology, Fondazione IRCCS Istituto Nazionale dei Tumori, 20133 Milan, Italy; maurizio.colecchia@isitutotumori.mi.it; 6Department of Oncology and Hemato-Oncology, University of Milan, 20133 Milan, Italy

**Keywords:** kidney cancer, renal cell carcinomas, collecting duct carcinoma, transcriptomic, molecular subtypes, gene expression, prognostic/predictive biomarkers, histological classification

## Abstract

**Simple Summary:**

The treatment of collecting duct carcinoma (CDC) remains unsatisfactory since this highly aggressive kidney cancer has an unfavorable clinical behavior. Due to the rarity of CDC a complete biological characterization of this disease is still lacking. The aim of our study is to provide new insights into the molecular biology of CDC leveraging newly generated and publicly available gene expression profiles of CDC tumors. We identified unique gene expression programs and pathways that distinguished CDC from other renal malignancies. The CDC-specific expression signature predicted in vitro sensitivity to several small molecule inhibitors screened in cancer cell lines from multiple cancer types. Finally, we proved that CDC is a molecularly heterogeneous disease made up of at least two subtypes distinguished by cell signaling, metabolic and immune-related alterations. Altogether, these findings pave the way for future investigations with meaningful clinical implications aimed at improving the management of CDC patients.

**Abstract:**

Collecting duct carcinoma (CDC) is a rare and highly aggressive kidney cancer subtype with poor prognosis and no standard treatments. To date, only a few studies have examined the transcriptomic portrait of CDC. Through integration of multiple datasets, we compared CDC to normal tissue, upper-tract urothelial carcinomas, and other renal cancers, including clear cell, papillary, and chromophobe histologies. Association between CDC gene expression signatures and in vitro drug sensitivity data was evaluated using the Cancer Therapeutic Response Portal, Genomics of Drug Sensitivity in Cancer datasets, and connectivity map. We identified a CDC-specific gene signature that predicted in vitro sensitivity to different targeted agents and was associated to worse outcome in clear cell renal cell carcinoma. We showed that CDC are transcriptionally related to the principal cells of the collecting ducts providing evidence that this tumor originates from this normal kidney cell type. Finally, we proved that CDC is a molecularly heterogeneous disease composed of at least two subtypes distinguished by cell signaling, metabolic and immune-related alterations. Our findings elucidate the molecular features of CDC providing novel biological and clinical insights. The identification of distinct CDC subtypes and their transcriptomic traits provides the rationale for patient stratification and alternative therapeutic approaches.

## 1. Introduction

Collecting ducts carcinoma (CDC) is a rare and aggressive histological subtype of renal cancer accounting for less than 1% of renal tumors. This malignant epithelial tumor arises from the distal segment of the collecting ducts of Bellini in the renal medulla, bearing distinct clinical, histological and pathological characteristics [1,2,3,4]. CDC has an extremely poor prognosis. About one third of patients present metastatic disease at diagnosis [5], and approximately two third of patients die within two years [6].

To date, the treatment of metastatic disease remains highly unsatisfactory and, given the rarity of CDC, only a few data are available to support the best therapeutic options for this aggressive malignancy. One of the major findings in terms of overall response rate was achieved in a prospective phase II trial showing the efficacy of the chemotherapy combination cisplatin plus gemcitabine in 23 previously untreated metastatic CDC (mCDC) patients [7]. Despite the relevant advantages of the use of targeted therapy in the treatment of clear-cell renal cell carcinoma (ccRCC), no results from prospective phase III trials evaluating these approaches in CDC are available. Again, only case reports of CDC patients treated with immune-checkpoint inhibitors are reported [8,9,10]. This gap in the clinical management of mCDC is certainly due to the rarity and the intrinsic aggressiveness of the disease, but the poor molecular characterization prevents the development of tailored treatments for mCDC management.

Only a few studies have examined the molecular profile of CDC. Targeted sequencing performed on 17 CDC cases led to the identification of recurrent somatic mutations in *NF2*, *SETD2* and *SMARCB1* genes with 29%, 24% and 18% frequency, respectively [11]. Another genomic profiling performed on seven CDC cases reported homozygous deletion of *CDKN2A* gene and a recurrent mutation in the *MLL* gene in about 50% of cases [12]. With the exception of *MLL* and *SMARCB1* mutations, all other somatic mutations were already reported in ccRCC [13]. Transcriptomic profiling through RNA sequencing (RNA-Seq), performed on seven CDC and four matched normal kidney samples, highlighted the dysregulation of several solute carrier family genes including tumor overexpression of *SLC7A11*, a cisplatin-resistance marker [12]. An independent study on two CDC and eight normal kidney samples reported an overexpression of cancer-related and cell cycle pathways in CDC, in addition to the modulation of SLC genes [14]. Another RNA-Seq study of 11 CDC and nine upper-tract urothelial carcinoma (UTUC) samples compared with additional renal cancer cell (RCC) histologies from the cancer genome atlas (TCGA) showed that CDC tumors clustered separately from other tumor types [15]. Furthermore, this study defined CDC as a metabolic disease characterized by a shift toward aerobic glycolysis and by an overexpression of immune genes related to lymphocyte activation and T cell proliferation [15]. Comparing bulk tumor RNA-Seq data to gene expression profiles of kidney cell types microdissected from different nephron sites, the authors provided indication that the transcriptional differences between CDC and other histologies may be linked to the cell type of origin of these tumors [15].

Even though these studies have provided useful knowledge about CDC biology, the small sample size of the analyzed cohorts hinders the general validity of their findings. Thus, the definition of pathognomonic transcriptional features of CDC is still lacking. In our study we applied an integrative analytical approach to robustly elucidate the molecular traits of this rare tumor type. Our results show that CDC tumors are characterized by a distinctive transcriptional program that recapitulates gene expression alterations linked to both cancer-related processes and cell of origin of the tumor. For the first time, we provide indication that CDC transcriptome may be predictive of sensitivity to several drugs in cancer cell lines, and we highlight the existence of at least two molecular subtypes of CDC driven by marked transcriptional differences.

## 2. Materials and Methods

### 2.1. Case Collection of the INT-CDC Cohort

Tumor samples from 6 CDC patients were collected at Fondazione IRCCS Istituto Nazionale dei Tumori of Milan (INT, Milan, Italy). Histological verification of specimens was performed by an expert uropathologist using established criteria for the diagnosis of CDC [16]. All patients underwent immunohistochemistry for INI-1 and fumarate hydratase. Positivity for high molecular weight cytokeratins (CK19 and 34βE12), CK7, and PAX8, with retained expression of SMARCB1 (INI1/BAF47) and fumarate hydratase (FH) supported the diagnosis of CDC. For comparison, tumor samples from 5 ccRCC patients and 4 samples from non-affected normal kidney tissue were collected. Normal samples were obtained from 3 ccRCC patients and 1 CDC patient. Patients’ information is reported in Table 1.

The collection of specimens and associated clinical data used in this study were approved by the Independent Ethics Committee of the Fondazione IRCCS Istituto Nazionale dei Tumori of Milan (Protocol number INT 31/14). The study was approved by the Institutional Review Board of The Ohio State University (Columbus, OH, USA) (Protocol number 2005C0014). All patients provided written inform consent to donating the tissue remaining after diagnostic procedures for analysis. Tissue collection and usage were performed in accordance with the ethical standards established in the 1964 Declaration of Helsinki and its later amendments. The privacy rights of human subjects have been observed.

### 2.2. RNA Extraction and Microarray Profiling

Total RNA was extracted from formalin-fixed, paraffin embedded samples using the RecoverAll Total Nucleic Acid Isolation Kit for FFPE (Invitrogen, Carlsbad, CA, USA), quantified on a Qubit fluorometer (Thermo Fisher, Waltham, MA, USA) and assessed for integrity on a 4200 TapeStation, (Agilent Technologies, Santa Clara, CA, USA). Gene expression profiles were generated using GeneChip Human Transcriptome Array 2.0 microarrays (Thermo Fisher). RNA labeling, processing, and hybridization were performed according to manufacturer’s instructions, and microarrays were scanned with the GeneChip System 3000 (Thermo Fisher) scanner. Quality controls were performed on raw and preprocessed data to control for batch effects and outliers.

### 2.3. Retrieval of Public Gene Expression Data

Additional transcriptomic data of CDC and normal samples were downloaded from the Gene Expression Omnibus (GEO) repository (raw RNA-Seq counts of GSE89122 [12] and raw microarray data of GSE11151 [17]). A third dataset, hereafter named Wach, was built from the raw RNA-Seq counts included in Appendix A of the corresponding publication [14].

RNA-Seq fastq files for renal medullary carcinoma (RMC) [18] were obtained from the NCBI sequence read archive (accession: PRJNA605003). TCGA RNA-Seq FPKM values, samples and patients annotations were obtained using the TCGABiolinks package [19]. Curated TCGA follow-up data were downloaded from the TCGA Pan-Cancer Clinical Data Resource dataset [20].

RNA-Seq raw counts for ccRCC patients included in the International Cancer Genome Consortium—Renal Cell Cancer—European Union/France (ICGC-RECA-EU) dataset were downloaded from the ICGC data portal [21] (accessed on 13 October 2020).

RNA-Seq RPKM values for UTUC samples included in the Cornell/Baylor/MDACC dataset (CBM-UTUC) [22] were downloaded from cBioPortal [23] (accessed on 13 October 2020).

For the Cancer Cell Line Encyclopedia (CCLE) [24], RNA-Seq RPKM data and cell lines annotation were downloaded from CCLE data portal (https://portals.broadinstitute.org/ccle (accessed on 14 October 2020)). Pharmacological data were obtained from the Appendix A of Cancer Therapeutic Response Portal version 2 (CTRP) main publication [25] (Appendix A). For cell lines included in the Genomics of Drug Sensitivity in Cancer (GDSC), raw microarray data were obtained from ArrayExpress repository [26] (accession number: E-MTAB-3610). GDSC drug response data, expressed as area under the dose–response curve (AUC), were obtained from Appendix A of Iorio et al. [27]. Cell lines and compounds annotations were obtained from Appendix A of Iorio et al. [27] and GDSC website (https://www.cancerrxgene.org/downloads/bulk_download (accessed on 14 October 2020)) [28], respectively. For all the remaining datasets, normalized gene expression data were downloaded from GEO repository. A complete list of the datasets analyzed including the number of tumor and normal samples is reported in Appendix A.

### 2.4. Data Preprocessing

Microarray raw data were preprocessed using appropriate methods for each specific platform. Raw Affymetrix CEL files were preprocessed using the frozen robust multi-array average (RMA) algorithm [29] from Bioconductor [30]. For INT and GDSC datasets, preprocessing was performed using the RMA [31] function implemented in the oligo [32] package. Illumina microarray data were log2-transformed and normalized using the robust spline method implemented in the lumi [33] package. For each dataset, platform annotation was obtained from dedicated Bioconductor annotation packages. Multiple probes mapping to the same gene symbol were collapsed using the collapseRows function of the WGCNA package with “MaxMean” method [34].

STAR [35] was applied to align fastq files of RMC dataset to the human genome (hg38) and obtain raw gene counts. RNA-Seq data were normalized using two methods. For differential expression analysis, raw counts were normalized for library size using the trimmed mean of M-value (TMM) method [36] implemented in the edgeR package [37]. For single sample gene set enrichment analyses, raw gene counts were normalized by the fragment per kilobase million (FPKM) calculation. Gene lengths were retrieved using the GOseq Bioconductor package. RNA-Seq datasets were annotated using GENCODE hg19 version 32 annotation and were collapsed at the gene level by summing the raw counts or FPKM/RPKM values of transcripts. Raw single cell (sc) RNA-Seq data of normal kidney cells (GSE131685) were processed using Seurat [38] and Harmony [39] as previously described [40].

### 2.5. Differential Expression Analysis

Differential expression analysis of microarray datasets was carried out using the standard linear modeling approach implemented in the limma package [41]. For RNA-Seq datasets GSE89122 [12] and Wach [14], TMM-normalized data were transformed in log2 counts per million with relative gene-level weights using the limma-voom approach [42]. Nominal *p*-values were corrected for multiple testing using the Benjamini–Hochberg false discovery rate (FDR) [43]. Differentially expressed genes were selected according to a |fold-change (FC)| ≥ 2 and an FDR < 0.05, except for INT dataset where a |FC| ≥ 2 and an FDR < 0.25 were applied. The 31 genes of the INT-CDC signature were identified by comparing CDC, ccRCC and normal samples of INT dataset with ANOVA using limma [41]. Genes with an FDR < 0.25, a FC ≥ 2 in the CDC versus ccRCC and CDC versus normal contrasts and a FC < 1.5 in ccRCC versus normal were selected.

For scRNA-Seq data, genes differentially expressed between cell types were identified using the FindMarkers function of the Seurat package [38] with Wilcoxon rank-sum test.

### 2.6. Functional Enrichment Analysis and Visualization

Over-representation analysis of gene lists was performed with the gProfiler2 package [44] with default parameters using the Gene Ontology Biological Process (GO:BP) terms and Reactome pathways. Significant terms were selected according to an adjusted *p*-value < 0.05 and were grouped in functional categories using the following network approach. We first measured pairwise gene content similarity between terms by computing the Cohen’s kappa statistic as previously described [45]. Term pairs with a kappa ≥ 0.5 were selected and imported in Cytoscape v3.8.1 for network generation [46]. Term clusters were identified using the gLay clustering algorithm [47] and manually annotated into functional categories. The igraph R package was used for network visualization. Over-representation analysis of gene families was performed using hypergeometric test. Gene families were retrieved from the HUGO gene nomenclature committee website (https://www.genenames.org (accessed on 4 November 2020)) [48]. Gene Set Enrichment Analysis (GSEA) [49] was performed in pre-ranked mode using the fgsea Bioconductor package. The C2 Reactome canonical pathways collection was retrieved from MSigDb database v 7.1 [50]. Genes were ranked using the *t*-statistic derived from the differential expression analysis with limma. The number of permutations was set to 10,000 and gene sets with less than 15 genes or more than 500 genes were filtered out. Significant gene sets were selected according to an FDR < 0.05. Reactome gene sets were manually grouped into functional categories according to the Reactome pathway hierarchy. Inference of immune cell infiltration from gene expression data was carried out using the ESTIMATE algorithm [51].

### 2.7. Meta-Analysis of Transcriptomic Datasets and Single-Sample Scoring

To compare CDC with other RCC histologies and normal kidney, we combined the INT-CDC dataset with all other datasets reported in Appendix A. Normalized microarray data and FPKM/RPKM RNA-Seq values were combined at the gene level. The final integrated dataset included expression values of 9152 genes for 2568 samples. Due to the presence of a confounding factor between the cancer histology and the source dataset, it was not possible to apply batch effect corrections. Instead we applied singscore, a true single sample scoring approach that produces stable scores across datasets independently from dataset composition and suitable for data integration [52]. Singscore was used with the INT-CDC signature or with C2 Reactome canonical pathways gene sets retrieved from MSigDb database v 7.1. Differentially expressed gene sets were identified using limma applied to the singscore values. Gene sets with significant (FDR < 0.05) up- or downregulation in all pairwise comparisons between CDC and other histologies were selected.

### 2.8. Cell of Origin Analysis

To compute similarity between bulk and single-cell expression data we first standardized expression values by calculating the z-score across samples for each gene. We then correlated bulk and single-cell expression data using Pearson’s correlation coefficient, after selection of the top 5000 most variant genes in single-cell data with the FindVariableFeatures of Seurat package [38]. Finally, for each bulk sample the median correlation coefficient across cells of each kidney cell type was calculated. To assess the enrichment of the INT-CDC signature in scRNA-Seq data we applied AUCell that allows the evaluation of gene sets activation in single cells [53]. Null distributions of AUCell scores were obtained by calculating scores for 1000 randomly generated gene sets of the same size of the INT-CDC signature. Distributions of AUCell scores between collecting duct principal cells and every other cell type were compared using Wilcoxon rank-sum test.

### 2.9. Unsupervised Analysis of CDC Tumors

To increase the number of CDC samples for unsupervised analysis, we first merged processed microarray data and log2(FPKM) values of the four datasets of CDC and normal kidney samples (INT, GSE89122 [12], GSE11151 [17] and Wach [14]). The final dataset included 18,048 genes for 38 samples (17 CDC and 21 normal kidneys). To reduce batch effects, genes were standardized across samples of each dataset by z-score calculation and the merged dataset was quantile-normalized. Unsupervised analysis was applied on the 17 CDC samples. A consensus partitioning of these tumors was identified using the COLA package that combines different clustering algorithms and gene variability metrics to identify robust subgroups. COLA was applied to increasing subsets of the data (top 1000, 2000, 3000, 4000 and 5000 most variant genes according to standard deviation, median absolute deviation, coefficient of variation and “ability to correlate” measure). Six clustering algorithms were applied (hierarchical, k-means, spheric k-means, partitioning around medoids, model-based clustering and non-negative matrix factorization). Given the low sample size, the number of tested clusters was limited from k = 2 to k = 4. Similarity between CDC clusters and TCGA ccRCC (TCGA-KIRC) and papillary RCC (TCGA-KIRP) molecular subtypes was assessed using the SubMap algorithm [54] implemented in GenePattern software [55]. Molecular classifications of TCGA samples were retrieved from the corresponding landmark publications [13,56] using the TCGAbiolinks package [19].

### 2.10. Prioritization of Anticancer Drugs

To identify candidate drugs for treatment of CDC we applied singscore [52] with the INT-CDC signature for each cell line in CCLE and GDSC. This score was then correlated to the AUC values of the drugs using Spearman’s correlation coefficient. Significant correlations were selected according to an FDR < 0.05. Differential expression between the 17 CDC and 21 samples of the merged dataset was performed using limma [41], to evaluate modulation of drug targets in tumor and normal samples. An FDR < 0.05 was applied to define significantly up- and downregulated genes.

The Connectivity Map database [57] (https://clue.io/ (accessed on 16 December 2020)) was interrogated using the list of 31 genes of the INT-CDC signature. From the results, we selected the connectivity scores for 10 drugs of interest, calculated for the 9 cancer cell lines (PC3, VCAP, A375, A549, HA1E, HCC515, HT29, MCF7, and HEPG2) profiled within the Touchstone database [57]. The mean connectivity score across the 9 cell lines was used to rank the compounds.

### 2.11. Survival Analysis

Survival analysis for kidney cancer patients of TCGA datasets was performed using the Kaplan–Meier curves method implemented in the survminer R package. Primary tumor samples were divided into a high and low category according to the median score of selected signatures, computed with the singscore package [52]. *p*-values were calculated using log-rank test. A *p*-value < 0.05 was considered statistically significant. Hazard ratios and confidence intervals were calculated using the Cox-proportional hazard model implemented in the survival package. The assumption of proportional hazards was evaluated by assessing the relationship between model residuals and time using the cox.zph function of survival R package.

## 3. Results

### 3.1. Transcriptional Divergence of CDC Compared to Normal Kidney and Definition of a Gene Signature Highly Expressed in CDC

To investigate the transcriptional alterations underlying CDC biology, we compared gene expression profiles of six CDC and four normal kidney samples included in the INT-CDC cohort. Results highlighted strong differences between the two groups of samples, with 227 and 72 genes significantly up- and downregulated, respectively (Figure 1A, Appendix A). The robustness of the alterations identified in the INT cohort was assessed by performing GSEA on three independent small datasets of CDC and normal kidney samples (Table 2).

The set of 227 genes upregulated in CDC cases of the INT-CDC cohort was significantly positively enriched in CDC samples of all three datasets (Figure 1B). As well, a negative enrichment was observed for the gene set of 72 genes found downregulated in the INT-CDC cohort (Figure 1B).

To identify genes robustly altered in CDC tumors in comparison to the normal counterpart, we intersected the lists of differentially expressed genes between CDC and normal renal tissue in the four analyzed datasets. A core of 469 up- and 569 downregulated genes occurring in at least three out of four datasets was identified (Figure 2A, Appendix A).

To gain a first categorization of these genes we analyzed them in relation to known gene groups and we identified significant over-representation of many gene families (Appendix A). We found highly significant over-representation of SLC genes, whose expression was previously shown to be altered in CDC [12]. Seven SLC genes, mainly encoding for amino acids and glycoprotein transporters, were upregulated in CDC (Appendix A). The 34 downregulated SLC genes were involved in transport of inorganic and organic cations and anions, bile salts, essential metals, neurotransmitters and drugs, nucleoside and fatty acids (Appendix A). Related to trans-membrane transport we additionally found over-representation of aquaporins that were all downregulated in CDC, chloride and potassium channel genes (Appendix A). To gain more insight into the biological function of these core genes, we performed functional over-representation analysis and we observed a significant enrichment of 444 and 145 GO:BP terms and Reactome pathways in the lists of core up- and downregulated genes, respectively (Appendix A). Using a network approach, significantly enriched pathways and biological processes were clustered into functional categories according to their similarity in terms of number of shared genes (Figure 2B). CDC showed upregulation of genes involved in cell cycle and proliferation, extracellular matrix, cell movement, and immune response. Downregulated genes were mainly involved in metabolism, in particular of carbohydrates and nucleotides, fructose, ketone bodies, as well as in the above mentioned SLC-mediated ion transmembrane transport (ion transport) and in development of renal structures (Figure 2B).

By comparing gene expression profiles of CDC samples to normal kidney and to five ccRCC samples included in the INT cohort, we identified a signature of 31 genes (hereafter named INT-CDC signature) specifically upregulated of at least two-fold in CDC compared to the other samples (Figure 3A, Appendix A). Of note, 23 (74%) of the 31 genes of the signature were robustly upregulated in CDC compared to normal samples (*p*-value < 0.05 and FDR < 0.25) in at least one of the three public CDC datasets (Appendix A). Furthermore, 13 (42%) of these genes were significantly upregulated (*p*-value < 0.05 and FDR < 0.25) in all three CDC datasets. This INT-CDC signature was enriched in genes involved in extracellular matrix organization, cell–cell communication, and epithelium development (Appendix A). We evaluated the overall expression of the INT-CDC signature at the single-sample level in public gene expression data of 2120 cancer samples encompassing additional CDC cases, ccRCC, papillary, chromophobe, oncocytoma, RMC, and UTUC. We confirmed that this gene signature was strongly expressed in CDC (Figure 3B). RMC showed high expression of the INT-CDC signature as well. Lower scores were observed in UTUC compared to CDC, followed by papillary carcinoma and all the remaining RCC histologies.

Given the poor prognosis of CDC, we hypothesized that the INT-CDC signature could be associated to high tumor aggressiveness. Due to the small sample size and the lack of follow-up data, the prognostic impact of the signature was not evaluable in CDC datasets. Therefore, we assessed the association of the INT-CDC signature with prognosis using transcriptomic and survival data of ccRCC, papillary, and chromophobe RCC patients of TCGA. Increased expression of the INT-CDC signature was significantly associated to poor overall survival and progression-free interval in ccRCC patients (Appendix A). No association with prognosis was observed in papillary and chromophobe RCC.

Among pathways significantly over-represented in CDC versus normal samples were those involved in immune response, immune signaling and interferon-γ signaling (Figure 2, Networks 13, 25 and 34). Recently, a T-cell inflamed tumor microenvironment signature, named Tumor Inflammation Score (TIS), was demonstrated to negatively correlate with metabolism in ccRCC and to be positively associated with response to anti-PD1 therapy in many cancer types, including renal cell tumors [58,59,60]. In comparison to all other histologies, with the exception of RMC and ccRCC, CDC significantly displayed higher immune infiltration score, as inferred by ESTIMATE algorithm (Appendix A) and higher levels of the TIS signature and other interferon-γ related gene sets (Appendix A). Overall, these findings highlight an enrichment of T cell infiltration in the tumor microenvironment of CDC, posing the conditions for some of these patients to benefit from immune checkpoint inhibitors.

### 3.2. CDC Transcriptional Program Defines Putative Active Drugs

Genes whose expression is modulated in tumor cells compared to the normal counterpart are useful for the screening of candidate therapeutic targets and for the development of novel drugs. To identify compounds that may be active in CDC we correlated the CDC transcriptional program with drug sensitivity in 1019 cell lines of CTRP and 971 cell lines of GDSC, for which both transcriptomic and pharmacological data are available. The CDC transcriptional program was defined by the score of the INT-CDC signature, calculated for each cell line using singscore. The INT-CDC signature score was finally correlated with drug response data, expressed as AUC, to identify positive and negative associations. Correlation analysis in CTRP dataset identified 28 compounds, among the 481 tested, which were significantly negatively correlated (FDR < 0.05) with the INT-CDC signature (Figure 4A, Appendix A). This means that a higher expression of the INT-CDC signature corresponds to lower AUC values, therefore to higher drug sensitivity. Of the 265 compounds assayed in GDSC, 74 showed a significant negative correlation with the INT-CDC signature (Figure 4A, Appendix A). The comparison of the Spearman’s correlation coefficients between the signature and AUC values of CTRP and GDSC showed overall agreement between the two datasets and led to the identification of 10 compounds negatively associated to the INT-CDC signature in both datasets (Figure 4B). The most negatively correlated drug was afatinib, a small molecule approved by the Food and Drug Administration (FDA) for the first-line treatment of non-small cell lung cancer with EGFR mutations. We found also three additional EGFR inhibitors: lapatinib, gefitinib, and erlotinib. The expression of the genes targeted by the 10 drugs was evaluated in CDC tumor samples in comparison to normal kidney (Figure 4C). For EGFR signaling inhibitors (erlotinib, afatinib, lapatinib, and gefitinib), we observed a significant upregulation of EGFR in CDC. SRC, a target of ABL signaling inhibitors (dasatinib, saracatinib, and bosutinib) was also upregulated, together with additional targets of dasatinib (EPHA2 and LCK). For the two MAPK inhibitors (selumetinib and trametinib) no differential expression of their targets (MAP2K1 and MAP2K2) was observed between tumor and normal tissue. Genes targeted by the insulin-like growth factor receptor inhibitor BMS-536924 were significantly downregulated in CDC.

Gene expression profiles of CTRP and GDSC cell lines were obtained from treatment-naïve cells, therefore all associations with drug sensitivity data were evaluable in a predictive setting. Assuming that the INT-CDC signature is a molecular proxy of a gene expression profile of a highly aggressive tumor type, we interrogated the Connectivity Map database to assess whether the 10 drugs negatively associated to the INT-CDC signature could actually revert its expression. The Connectivity Map provides positive scores if the perturbational signatures in the database are positively correlated with the query signature. Negative Connectivity Map scores indicate dissimilarity between perturbational and query signatures. All drugs except for bosutinib and lapatinib showed negative scores, supporting the evidence that the perturbational gene expression signatures of these drugs are negatively correlated with the INT-CDC signature (Figure 4D). Therefore, most of these treatments are likely to revert the CDC transcriptional program.

### 3.3. CDC Arises from the Principal Cells of Collecting Ducts

CDC has a presumed origin from the principal cells of the distal collecting ducts of Bellini. To confirm this hypothesis from a molecular perspective we compared, at single-cell level, the expression profiles of bulk CDC samples and other renal cancer histologies to scRNA-Seq data obtained from normal kidney [40]. Correlation analysis of bulk tumor and scRNA-Seq data showed that CDC were more similar to the principal cells of the collecting duct than to any other cell type of normal kidney (Figure 5A). RMC and UTUC showed the highest correlation with principal cells of the collecting duct as well, confirming the transcriptional similarity of CDC to these two malignancies. Papillary RCC showed higher similarity with glomerular parietal epithelial cells, while ccRCC showed overall poor correlation with all kidney cell types. Expression profiles of chromophobe renal cell carcinoma and oncocytoma were both more correlated to intercalated cells of the collecting duct. We next calculated the activation state of the INT-CDC signature in each cell type of the scRNA-Seq normal kidney dataset using the AUCell method. The population of collecting duct principal cells was characterized by significantly higher AUCell scores of the INT-CDC signature compared to all other cell types (Figure 5B). The analysis was repeated using 1000 random gene sets of the same size of the INT-CDC signature. Random gene sets had low AUCell scores in all cell types, indicating poor activation. No differences across cell types were observed. To exclude that the INT-CDC signature merely recapitulates the different cell lineage of CDC, we investigated the expression of the signature genes in scRNA-Seq data of the normal kidney. Results showed that only six genes (S100A11, KRT19, SLPI, SPINT2, PERP, and CDH1) were significantly upregulated in the principal cells of the collecting duct compared to other cell types (Appendix A). Overall, these results confirm from a molecular point of view that CDC arises from malignant transformation of the principal cells of the collecting ducts. The INT-CDC signature includes both cancer-related and lineage-specific signals and its upregulation in CDC and RMC is linked to their common cell type of origin.

### 3.4. CDC Shows Unique Pathway Enrichments

To obtain a broader picture of the specific transcriptional features underlying CDC biology, we compared expression profiles of CDC to other renal cancer histologies (ccRCC, papillary, chromophobe, oncocytoma, RMC), UTUC, and normal kidney. To reduce the heterogeneity of the different datasets related to sample composition and profiling technology, the comparison was performed at the level of single-sample scores for Reactome pathways. We observed the highest number of differentially expressed gene sets between CDC and chromophobe tumors (Figure 6A). The comparison between CDC and RMC gave the lowest number of differentially expressed gene sets (Figure 6A). We next selected gene sets that showed a consistent up- or downregulation in CDC compared to each other class in a pairwise manner (Appendix A). We found nine gene sets specifically downregulated in CDC, related to biological oxidations, amino acid metabolism, cell–cell communication, cellular response to external stimuli, and GPCR signaling (Figure 6B). On the other hand, CDC showed specific upregulation of 11 gene sets involved in DNA repair, organelle biosynthesis, innate immune system, interleukin-37, and Rho-GTPases signaling (Figure 6B).

### 3.5. CDC Shows Intertumor Heterogeneity

To assess whether CDC comprises distinct molecular subtypes, we applied unsupervised analysis to gene expression profiles of all the available CDC samples (*n* = 17). Given the low sample size and the difficulty in identifying robust clusters we applied COLA, an approach that enables the identification of consensus clusters combining different gene variability measures and partitioning algorithms. The tested 24 methodological combinations led to the identification of two major consensus clusters, CDC-S1 and CDC-S2 (Appendix A). We then compared the gene expression profiles of CDC-S1 (*n* = 8) and CDC-S2 (*n* = 9) to explore the molecular differences underlying these two subtypes. The INT-CDC signature was equally expressed in the two clusters, indicating that its expression is not affected by intertumor heterogeneity (Appendix A). CDC-S1 showed a positive enrichment of gene sets linked to extracellular matrix, muscle contraction, and neuronal system (Figure 7A, Appendix A). These last two categories include many genes involved in ion transport across membranes and maintenance of membrane potential (Appendix A). CDC-S2 instead displayed enrichment of gene sets related to multiple metabolic pathways, cell cycle, DNA repair, gene expression, programmed cell death, extracellular matrix organization, and vesicle-mediated transport (Figure 7A, Appendix A). Among gene sets related to small molecule transport, ion channels and SLC-mediated transport were enriched in CDC-S1 whereas CDC-S2 had increased uptake and transport of iron and increased transport mediated by ABC family proteins (Figure 7A, Appendix A). CDC-S1 cluster also displayed a prominent enrichment of signaling pathway mediated by G-protein coupled receptors (GPCR), whereas CDC-S2 had enrichment of WNT, NOTCH, Hedgehog, RTKs (EGFR, ERBB2, FGFR, KIT and VEGF), MAP kinases, and Rho-GTPases signaling (Figure 7A, Appendix A). Pathways belonging to innate immune system, cytokine and Rho GTPase signaling, DNA repair, organelle biogenesis and maintenance displayed higher expression in CDC-S2 subtype (Appendix A). Finally, the two clusters differed in terms of expression of immune system genes. Innate immunity linked to antimicrobial activity displayed a positive enrichment in CDC-S1, conversely several components of innate and adaptive immunity were depleted in this subtype. Adaptive immune-related pathways included T-cell receptor-, B-cell receptor-, and cytokine-signaling as well as MHC-class I and MHC-class II antigen processing and presentation, all enriched in CDC-S2 (Appendix A). No differences for overall immune infiltration, TIS and interferon-γ gene sets were observed between the two CDC subtypes (Appendix A).

Comparing CDC subtypes with transcriptional subtypes previously identified by TCGA-KIRC and TCGA-KIRP, we found that CDC-S1 did not display significant similarity to any of them, except CDC-S2 which showed a transcriptional agreement with m1 subtype of TCGA-KIRC (Figure 7B). According to previous findings, m1 TCGA-KIRC subtype is characterized by better prognosis compared to other molecular subtypes [13]. Similarly, ccRCC patients showing higher expression of the top 150 upregulated genes in CDC-S2 had better OS and progression-free interval (Figure 7C) suggesting that CDC tumors belonging to CDC-S2 subtype may be less aggressive than their counterpart CDC-S1.

## 4. Discussion

CDC is a highly aggressive subtype of renal cancer whose molecular determinants are still poorly characterized due to the rarity of cases and the complexity of diagnosis. In one large multi-institutional cohort, that included our own Institution with an expert pathologist (M.C.), 25% of cases diagnosed previously as CDC were recently reclassified as FH-deficient RCC [61]. A similar rate of CDC reclassification was found in a sequencing study [11]. All the tumors described in our cohort have received diagnosis of CDC according to microscopic morphology and immunohistochemistry to support the diagnosis of CDC and exclusion of additional entities such as other types of RCC, metastatic carcinoma, and urothelial carcinoma, including BK polyomavirus-associated upper tract urothelial carcinoma as described [62]. In our study, we applied an integrative analytical approach to transcriptomic data to investigate under different aspects the transcriptional landscape of CDC. We identified the genes and pathways distinguishing CDC from normal kidney, UTUC and other RCC histologies and we showed that the transcriptional program of CDC predicts sensitivity to specific drugs using in silico pharmacogenomic data. In addition, we molecularly confirmed that CDC originates from the principal cells of the collecting duct by comparing expression profiles of bulk tumors and single cells of the normal kidney. Finally, for the first time, we provided evidence that CDC is a heterogeneous disease that may comprise different molecular subtypes.

Previous studies, all including only from two to seven CDC cases, identified a large number of genes differentially expressed between CDC and normal kidney but none of them evaluated the validity of their findings in independent datasets [12,14,15]. Here, by making use of all the publicly available datasets of CDC and normal kidney samples, we were able to identify transcriptional differences that were reliable and reproducible across independent studies. This allowed us to confirm the dysregulation of the SLC transporters family, membrane-bound proteins involved in many physiological processes regulating ion, metabolite, and other solute exchanges, as well as cellular metabolism [63,64,65,66]. Transporters for essential amino-acids and nutrients often serve as oncogenes when upregulated [66]. This modulation allows cancer cells to reprogram their metabolic pathways in order to support increased biosynthetic and bio-energetic demands [66]. Among the most upregulated SLC genes, we found two amino acid transporters of the SLC7 family, the leucine preferring transporter SLC7A5 (also known as LAT1) and the cystine/glutamate antiporter SLC7A11 (also known as xCT). SLC7A5 is highly expressed in most cancers including RCC where its expression levels inversely associate to OS and progression-free survival [67]. Moreover, its selective inhibition in two RCC cell lines suppressed tumor growth and invasion in a dose-dependent manner making it a potential drug target [67]. Overexpression of SLC7A11 was also associated with poor outcome in ccRCC, chromophobe, and papillary RCC patients in the TCGA dataset [12]. In addition, SLC7A11 was reported to promote cystine uptake and glutathione biosynthesis, resulting in protection from oxidative stress and ferroptosis [68], a form of controlled iron-dependent cell death that is attracting increasing attention as a new option for cancer therapy. Cell-surface transporters like SLC7A5 and SLC7A11, can therefore be envisaged as potential novel therapeutic targets. Treatment options including drugs targeting the solute carriers upregulated in CDC have been previously suggested [63,64].

Conversely, in our analyses downregulated SLC genes could potentially play a role as tumor suppressors [69]. Besides SLC family genes, other components of small molecule transport like aquaporins and ion channels as well as processes involved in chemical, ion, and cation homeostasis are overall impaired in CDC compared to normal kidney. Among downregulated pathways we identified also many metabolic processes including carbohydrates and fatty acids metabolism, as already reported [15]. Our results additionally highlight a downregulation of fructose, amino acids, nucleotides, and ketone body metabolism, reinforcing the idea that CDC is a metabolic disease [15]. Our analyses also confirm previous observations that defense, immune and innate response, as well as proliferation and cell cycle are activated in CDC [15]. These findings, together with the upregulation of cell movement, response to wounding and extracellular matrix organization, indicate that CDC is a highly proliferative tumor with the ability to remodel the extracellular matrix and invade surrounding tissue. These features may also explain its metastatic potential and aggressiveness.

Previous CDC gene expression studies were limited to the comparison between tumor and normal tissue or UTUC samples. For the first time, we carried out a comprehensive comparison of CDC and all the other RCC histologies and UTUC to identify pathognomonic features of this tumor type.

We identified a signature of 31 CDC-specific genes (the INT-CDC signature) whose expression was higher in CDC than other RCC histologies and UTUC, except for RMC that showed comparable levels with CDC. CDC and other RCC histologies and UTUC were found to be distinct tumor entities also at the pathway level, but less marked differences were highlighted between CDC and RMC. Future investigations will require orthogonal approaches, such as real-time qPCR or immunohistochemistry, in larger cohorts that include CDC and other kidney cancer to further explore the specificity of the INT-CDC signature as well as similarities with RMC. Such validation will ultimately lead to its refinement and to the development of a molecular diagnostic tool that may help clinicians to identify CDC. CDC and RMC were reported to be similar also in terms of clinical, histological features and anatomical site of origin [3,70]. Our data provide evidence of commonalities also at the molecular level. Indeed, their expression program recalls that of the principal cells of the collecting duct of normal kidney, prompting a common cell of origin for these two kidney cancer types. Malouf and colleagues already suggested that CDC originates from the cells of the distal collecting duct [15]. However, the dataset of microdissected nephron cells [71] used to infer the cell of origin included only one biological replicate for each cell type, thus hindering analytical robustness. Since the INT-CDC signature included both genes upregulated in tumor versus normal tissue and genes differentially expressed between kidney cell types, we reasoned that it is a proxy of both cancer-associated and lineage-specific signals and its upregulation in CDC and RMC is linked to their common cell type of origin.

The identification of processes uniquely enriched or depleted in CDC might also provide therapeutic options for patients. CDC-specific upregulated features included DNA repair, innate immune system, and Rho GTPases. Rho GTPases are master regulators of actomyosin structure and dynamics and play pivotal roles in a variety of cellular processes including cell morphology, motility, adhesion, metabolism, survival, proliferation and cell cycle, stress, inflammation, gene transcription, and apoptosis [72]. Small-molecule inhibitors targeting Rho GTPase signaling may add new treatment options for cancer therapy, alone or in combination with other anticancer agents [72].

Through the analysis of cancer cell line pharmacogenomics datasets, we showed that the INT-CDC signature was also predictive of higher sensitivity to tyrosine kinase inhibitors (TKIs), including EGFR, ABL, and IGFR signaling inhibitors, and to SRC and MAPK signaling inhibitors. Assessment of the translational value of such in silico drug sensitivity predictions will require derivation of well characterized CDC patients’ cell lines or primary cultures to be used for preclinical screening of anticancer drugs.

In vitro, the combination of saracatinib, a SRC kinase inhibitor, and the TKI sunitinib, was proven to inhibit proliferation and migration in RCCs [73]. Metastatic RCC cells with wild-type VHL have hyperactivated SRC signaling and are sensitive to dasatinib treatment [74]. Due to its rarity, mCDC has often been excluded from randomized phase II-III trials evaluating targeted therapies in kidney neoplasms and until recently clinical evidence of activity of TKIs (such as sorafenib, cabozantinib, sunitinib, pazopanib) or other targeted agents (mTOR-inhibitors and anti-Her2 agents) were mostly limited to case reports or retrospective series [75,76,77,78]. Procopio et al. showed activity and efficacy and a favorable safety profile of a wide spectrum of TKIs and mTOR pathway inhibitors as upfront or subsequent line treatment in a retrospective study involving 13 mCDC patients [79]. Recently, a prospective, multicenter, single-arm, open-label, phase 2 trial evaluating the combination of sorafenib and chemotherapy with cisplatin plus gemcitabine, reported an ORR of 30.8% and a disease control rate (DCR) of 84.6% in previously untreated mCDC patients [80]. Tannir et al. reported the outcomes of a phase II study of sunitinib in patients with advanced non-clear cell RCC [81]. Among the six patients with mCDC or medullary carcinoma, no objectives responses were observed, whereas four patients achieved disease stabilization and two patients experienced disease progression [81]. Even though direct trial-to-trial comparisons cannot be carried out, the outcomes of patients with CDC treated with targeted therapies in the aforementioned studies are highly heterogeneous, highlighting the unmet clinical need of an adequate molecular knowledge of this disease in order to perform tailored biology-driven treatment choices.

As far as immunotherapy is concerned there is anecdotal evidence for the efficacy of checkpoint inhibitors in CDC patients [8,9,10]. By exploring the distribution of gene expression-based signatures associated to clinical response to anti-PD1 monotherapy in multiple cancer types [58,59,60] we found that CDC, RMC, and ccRCC have the highest enrichment across all histologies. Altogether, these results are encouraging and suggest that immune checkpoint inhibitors might be considered as promising therapeutic options. However, enrichment of the TIS and interferon-γ signaling signatures are only a side of determinants that might be critical for response to occur. Indeed, a recent study that considered whole-exome sequencing and transcriptomic data tumors of over 1000 patients treated with immune checkpoint inhibitors, identified in a multiparametric biomarker, that include features of host immune system together with antigenicity and genomic aberrations of the tumor, the best predictor to determine tumor response to immunotherapy [82]. Therefore, further work will be required to identify CDC patients more likely to benefit from this treatment option.

Heterogeneous treatment responses can be related to molecular heterogeneity. Our data give the first indication that at least two CDC subtypes exist. Even if some of the cases retrieved from the literature may have been misdiagnosed, the equal distribution of samples of the INT-cohort in the two clusters indicates that these subtypes are not related to the presence of different tumor types. The two subtypes showed large differences in many biological pathways including metabolism, cell cycle, DNA repair, signaling pathways, and tumor immune microenvironment. Classification of CDC in molecular subtypes may have prognostic and predictive implications. We found that the CDC-S2 subtype is similar to the TCGA-KIRC m1 subtype that was associated to better survival in ccRCC patients [13]. CDC-S1 subtype displayed upregulation of GPCR-mediated signaling whereas several signaling pathways, including those mediated by RTKs (EGFR, ERBB2 and VEGFR2) and downstream cascades (MAPK, AKT, mTOR) were upregulated in CDC-S2. The two clusters also differed in terms of their tumor immune microenvironment, although no differences were observed for the TIS and interferon-γ signatures. Adaptive and innate immune processes such as MHC-class I and MHC-class II antigen processing and presentation and those mediated by T-cell receptor, B-cell receptor, cytokines, and toll like receptors were all enriched in CDC-S2. Based on these findings we can hypothesize a different response among subtypes to currently available and novel treatments with the CDC-S2 subtype more likely to benefit of targeted therapies with TKIs.

## 5. Conclusions

Overall, this study expands the previous knowledge and provide novel insights into the biological processes that define this rare RCC histology. Although we collected and integrated all CDC expression data publicly available so far to increase robustness and statistical power, we recognize that the sample size is still limited. In addition, we are aware that analysis of data from multiple sources and platforms introduces batch effects that may have a detrimental impact on the results by masking important signals in the data. Despite such limitations, we identified, for the first time, significant pathways discriminating CDC tumors not only from normal cells but even from other RCC cancer histologies, derived a gene signature upregulated in CDC that allowed in-silico predictions for new therapeutic options, discussed the role of overexpressed molecular targets and key signaling pathways as new therapeutic targets. We also provide the first evidence for the existence of two molecular subtypes of CDC guided by different gene signatures, whose validation in larger homogeneous datasets might lead to more tailored and effective therapies for this cancer type with urgent unmet clinical need.

## Figures and Tables

**Figure 1 cancers-13-02903-f001:**
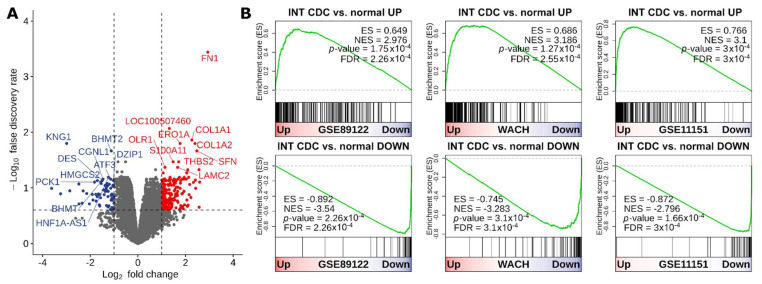
Differential gene expression between CDC and normal kidney. (**A**) Volcano plot showing differentially expressed genes between CDC and normal samples in INT dataset. The *x*-axis shows the log2 fold change. The *y*-axis shows the −log10 of the false discovery rate. An absolute log2 fold-change ≥ 1 and an FDR < 0.25, represented by the vertical and horizontal dashed lines, respectively, were used to select differentially expressed genes. Up- and downregulated genes in CDC are highlighted in red and blue, respectively. The top-10 up- and downregulated genes are reported. (**B**) Enrichment plots from GSEA conducted with the INT signature of up- and downregulated genes in CDC in three independent datasets of CDC and normal kidney samples. The red-to-blue color bar shows the ranking of the genes of each dataset from up- to downregulated in CDC. The vertical black bars indicate the position of the genes in the INT signatures along the ranked gene list. The green line shows the running enrichment score (ES) along the ranked gene list. NES: normalized enrichment score.

**Figure 2 cancers-13-02903-f002:**
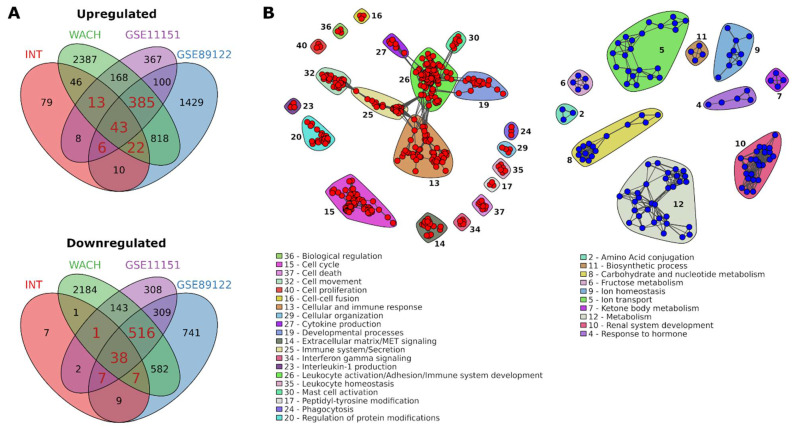
Validation and functional annotation of CDC deregulated genes. (**A**) Intersection of the genes significantly up- (upper panel) and downregulated (lower panel) in CDC versus normal kidney in INT cohort and three additional independent datasets. Genes found in at least three out of four datasets were highlighted in red; (**B**) network of pathways significantly over-represented (FDR < 0.05) in the list of validated differentially expressed genes. Red and blue nodes represent pathways significantly enriched in CDC and normal samples, respectively. Clusters of interconnected nodes identify pathways with genes in common above a Cohen’s kappa statistic of 0.35 and linked to the same biological process.

**Figure 3 cancers-13-02903-f003:**
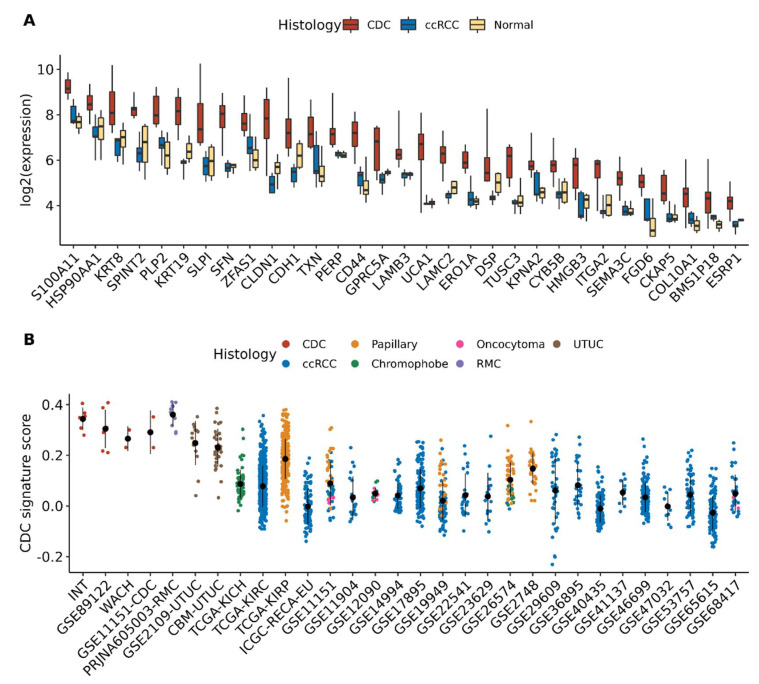
CDC-specific gene signature. (**A**) Boxplot of 31 genes specifically upregulated in CDC compared to normal and clear cell RCC samples in INT dataset; (**B**) distribution of the single-sample enrichment scores of the CDC-specific INT signature in transcriptomic datasets of different kidney cancer histologies. The black dot and line represent the mean and standard deviation of the score in each dataset.

**Figure 4 cancers-13-02903-f004:**
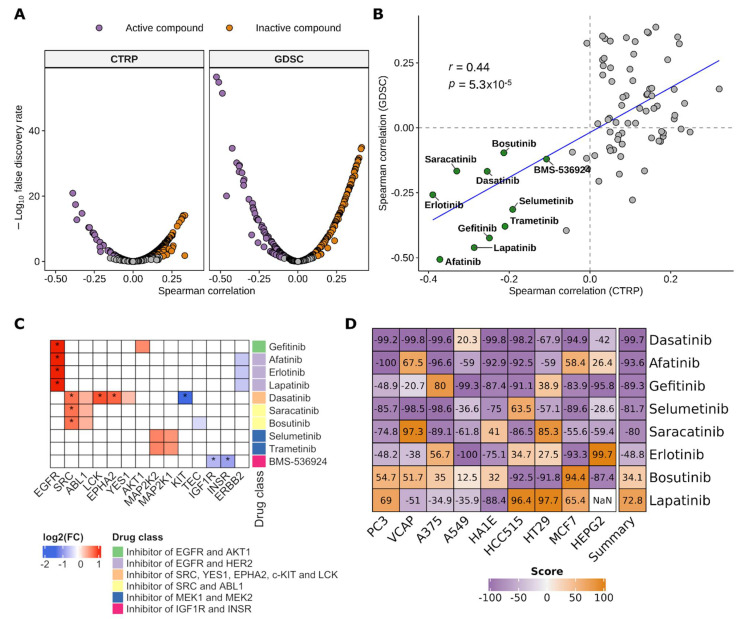
Predicted candidate drugs for CDC treatment. (**A**) Volcano plot showing the results of correlation analysis between the INT-CDC signature scores and drugs AUC values across cell lines of CTRP and GDSC datasets. Active compounds: negative correlation with FDR < 0.05; inactive compounds: positive correlation with FDR < 0.05. (**B**) Correlation between CTRP and GDSC Spearman’s correlation coefficients. Active compounds identified in both datasets are highlighted in green. Active compounds identified in GDSC only are highlighted in light blue. The blue line represents the regression line of the values. (**C**) Heatmap showing the modulation of target genes in the comparison between 17 CDC and 21 normal kidney samples. Up- or downregulation were defined according to an FDR < 0.05. (**D**) Heatmap showing the Connectivity Map scores of selected drugs in the nine cell lines profiled in the Touchstone dataset. The last column shows the mean score across the nine cell lines. Tissue of origin of human cancer cell lines: PC3: prostate; VCAP: prostate; A375: melanoma; A549: lung; HA1E: kidney; HCC515: lung; HT29: colon; MCF7: breast; HEPG2: liver.

**Figure 5 cancers-13-02903-f005:**
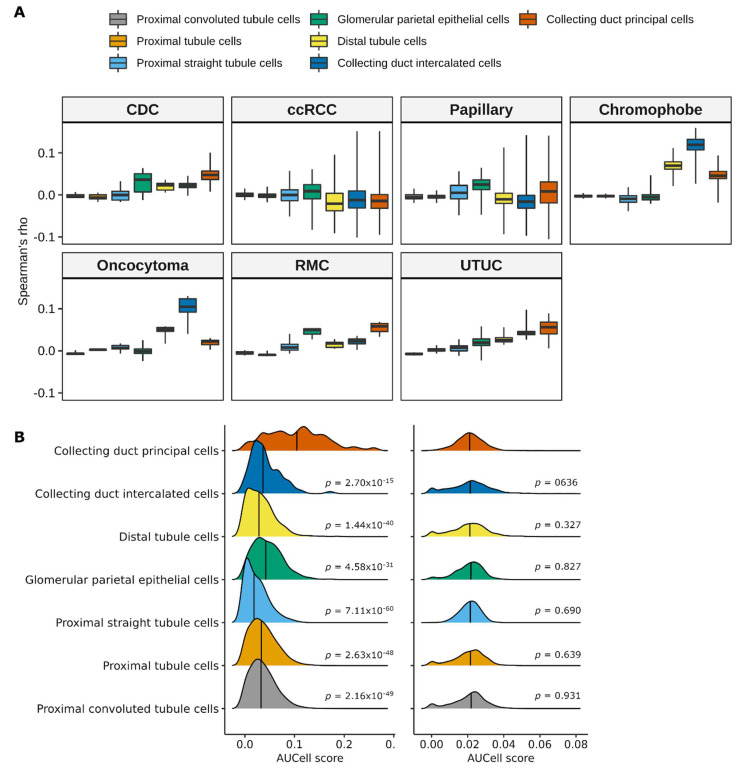
Cell-of-origin of CDC tumors. (**A**) Boxplot showing the correlation between bulk gene expression profiles of different kidney tumor histologies and single-cell transcriptomics data of normal kidney cell types from dataset GSE131685; (**B**) distribution of the single-sample AUCell scores of the CDC-specific INT signature (left column) in each kidney cell type from dataset GSE131685. Null distribution of 1000 random gene sets of the same size of the CDC-specific INT signature is reported in the right column. *p*-values by Wicoxon rank-sum test between each cell type and collecting duct principal cells.

**Figure 6 cancers-13-02903-f006:**
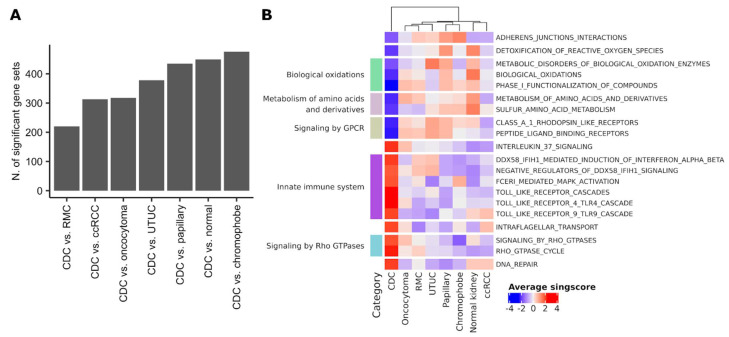
CDC-specific pathways modulation. (**A**) Number of differentially expressed Reactome gene sets (FDR < 0.05) between CDC and other RCC histologies; (**B**) heatmap showing Reactome gene sets significantly up- or downregulated in CDC compared to all other kidney cancer histologies and normal kidney. Functional categories defined by the Reactome hierarchy are reported on the right.

**Figure 7 cancers-13-02903-f007:**
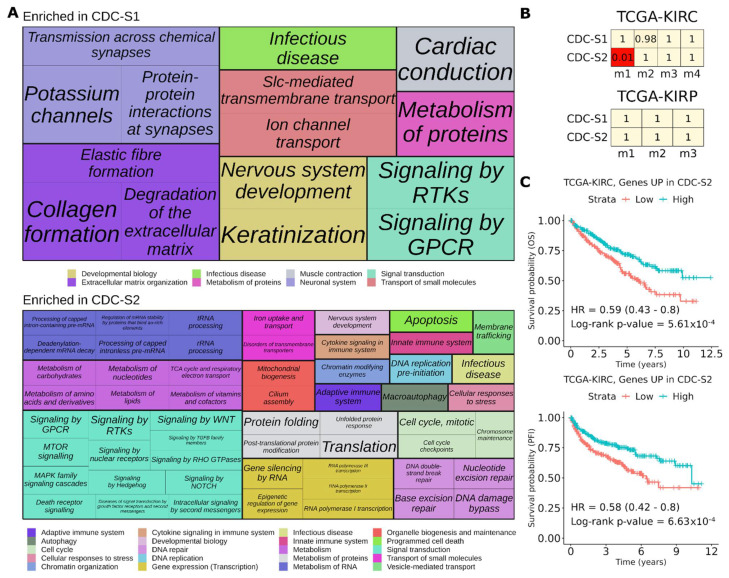
Functional analysis of CDC subtypes. (**A**) Treemap showing functional categories of Reactome gene sets significantly (FDR < 0.05) differentially enriched in CDC-S1 versus CDC-S2 subtypes. Gene sets were grouped according to the Reactome pathway hierarchy, and they are highlighted by different colors. (**B**) Bonferroni-adjusted *p*-values of the similarity between CDC and TCGA ccRCC (KIRC) or papillary (KIRP) subtypes. An adjusted *p*-value < 0.05 indicates significant similarity. (**C**) Kaplan–Meier curves referred to overall survival (OS, upper panel) and progression free-interval (PFI, lower panel) of TCGA ccRCC patients stratified according to the median expression of the top 150 genes upregulated in CDC-S2 subtype. HR: hazard ratio.

**Table 1 cancers-13-02903-t001:** Patient characteristics of the INT cohort.

Patient ID	Age	Sex	Histology	Matched Normal	Nephrectomy	Tumor Specimen Site	Treatment-Naïve Specimen	Vital Status	Overall Survival (Months) *
PG-2	67	F	CDC	No	Yes	Metastasis (soft tissue)	No	Deceased	54.6
PG-3	64	F	CDC	Yes	Yes	Primary tumor	Yes	Deceased	3.3
PG-4	43	M	CDC	No	Yes	Primary tumor	Yes	Deceased	7.7
PG-5	36	F	ccRCC	No	Yes	Primary tumor	Yes	Alive	228.1
PG-6	43	M	CDC	No	No	Primary tumor	Yes	Deceased	7.9
PG-7	57	M	ccRCC	Yes	Yes	Primary tumor	Yes	Deceased	7.4
PG-8	71	M	ccRCC	Yes	Yes	Primary tumor	Yes	Deceased	47.2
PG-9	33	F	ccRCC	Yes	Yes	Primary tumor	Yes	Deceased	58.3
PG-14	71	F	ccRCC	No	Yes	Primary tumor	Yes	Alive	40.9
PG-15	46	F	CDC	No	Yes	Primary tumor	Yes	Deceased	6.7
PG-16	36	F	CDC	No	Yes	Primary tumor	Yes	Deceased	24.0

* For alive patients the time from diagnosis to last follow-up is reported.

**Table 2 cancers-13-02903-t002:** List of CDC gene expression datasets analyzed in the study.

Authors	Study Label	Reference	CDC (n)	ccRCC (n)	Chromophobe (n)	Oncocytoma (n)	Papillary (n)	Normal Kidney (n)
Gargiuli C et al.	INT-CDC	This paper	6	5	0	0	0	4
Wang J et al.	GSE89122	[12]	7	0	0	0	0	6
Yusenko MV et al.	GSE11151	[17]	2	26	4	4	19	5
Wach S et al.	WACH	[14]	2	0	0	0	0	8

## Data Availability

Raw and preprocessed data were deposited in the GEO database repository with accession number GSE153965. All scripts to reproduce main figures and Appendix A are available at https://github.com/mdugo/CDC_transcriptomics_ms.

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
