# Peer review of "Integrative Transcriptomic Analysis Reveals Distinctive Molecular Traits and Novel Subtypes of Collecting Duct Carcinoma"

_cancers, 2021, doi:10.3390/cancers13122903_

Round 1

Reviewer 1 Report

Gargiuli et al presents a highly innovative and complex bioinformatic delineation of CDC transcriptomic data that reveals a unique gene signature for CDC as well as confirms its CD principal cell origin, and for the first time identifies two CDC distinct subtypes and predicts possible drug responses.  Overall, the manuscripts contains sophisticated bioinformatic approaches that provide a unique and insightful understanding to the transcriptional landscape of CDC and may be helpful to inform future diagnostic and treatment approaches. Although the basis for the study is microarray data, which could be argued to be an outdated transcriptomic methodology the obtained results are of significant value.

Comments:

  1. The authors identify a CDC specific gene signature in their INT cohort (INT-CDC) of 31 genes. It would be of interest to show e.g. in a venn diagram if these 31 genes are commonly found to be differentially expressed between CDC and normal in the WATCH, GSE11151, and GSE89122 data. Also, I find it unfortunate/under-powered that the authors chose to only use the INT cohort to generate the signature given that other cancer types and CDC data sets were available. I would thing a fine-tuning of that signature would have been possible.
  2. The survival analyses as defined by INT-CDC gene signature expression is truly interesting. I appreciate the rareness of CDC and making it impossible to use the signature for survival analysis of this cancer type. But would enough samples be available if combining survival data of CDC, RMC and UTUC, given that they have the most similar score of the signature genes?
  3. I personally find the results depicted in Figure 4C difficult to understand as the figure is. A similar heatmap representation as in 4D would be more informative.
  4. It is unclear whether for the pathway enrichment or the inter-tumor heterogeneity only the data from the INT cohort was used. Could the data be strengthened including WATCH, GSE11151, and GSE89122 data?
  5. Are the two subtypes of CDC different enough that if using all available data sets it is possible to derive a unique signature and drug response for each?
  6. The authors state that it is at times difficult to diagnose CDC correctly. Can the signature be used to correctly ID CDC from different cancer types based on transcriptomic data? Meaning, in the setting of personalized medicine, if transcriptomic data is available from a renal cancer patient tumor sample, can the signature be used for diagnosis?
  7. The paper lacks validation of their bioinformatic findings on CDC tissue samples (gene signature by IHC or IF, or immune cell infiltration by IHC or IF) as well as drug sensitivity using cell lines/primary cells.

Author Response

Please see the attachment. The Reviewers’ comments are shown in italics followed by our response. Sentences added to the manuscript are underlined.

Reviewer 2 Report

The authors have prepared an interesting study analyzing gene expression seen in a cohort of rare tumors, collecting duct carcinoma, as compared to normal kidney and a few additional clear cell renal cell carcinomas. The numbers are low, limited by rareness of this tumor as well as the diagnosis being one of rigorous exclusion by contemporary diagnostic criteria. However, they were able to extract distinctive gene expression signatures, which they the assayed across data from several previously published collecting duct carcinoma cohorts for consistency and associations of interest.  The conclusions are not exceptionally groundbreaking, but they do provide important supportive evidence regarding gene expression relating these tumors to the principal cells of the collecting duct through comparison of their signature to very recent single cell based sequencing studies of specific cells within the kidney.  Additionally, a hypothesis generating analysis comparing their collecting duct carcinoma signature to databases of cell lines with in vitro sensitivity and gene expression data nominates several specific drugs and classes.  Overall strengths include leveraging several published cohorts to supplement the overall weakness, that of a relatively few cases analyzed.  Interpretation and impact of this work would be improved as follows:

1) Diagnosis of CDC is one of exclusion, while the line delineating CDC from unclassified tumors is somewhat arbitrary.  Since only six (new) cases were studied, inclusion of a figure (supplementary or primary) showing their representative histology would be of use, along with richer clinical/pathologic data (grade, tumor size, stage, sex) and any available follow up. 

2) While Supplementary Table S1 provides full data on all of the cohorts and counts of different tissue types analyzed, including a smaller table in the primary document providing the study "label" (e.g. WACH), counts of tumors/tissue types analyzed, and references to published literature for the main cohorts studied would be useful to the reader to reference.  

3) Perhaps the most interesting aspect of the findings described concerns identification of heterogeneity within the cohort of 17 CDC samples that they assemble by zcore normalizing across cohorts.  At a minimum, a table should be included in the supplement providing all available details on each of members of these two subgroups, indicating from which study they originated, and any other known clinical features. Can an unsupervised cluster of the whole cohort be presented before embarking on the COLA approach? Does the CDC-S1 versus S2 distinction break down along any other interesting clinicopathologic lines? Is there a way to visualize these data that is more intuitive than what is currently shown in figure 7? Similarly, the association of gene expression distinctive to cluster S2 with a subtype of good prognosis clear cell tumors seems like a strange non sequitur (given the extrapolation across histologic subtypes of renal epithelial neoplasia). 

4) Have the authors deposited their original cel file and processed data into a public database? How about data their merged normalized cohort of the 17 cases and controls?

Author Response

(The authors gave the same response as above.)

Round 2

Reviewer 1 Report

I thank the authors for their detailed responses. I have only one minor comment. I would suggest adding to the discussion the author response to my last comment – in vivo validation of the findings. I think it is worth while to have a discussion section of future directions that encumbers the notion of validating the identified signature by IHC, as well as in larger cohorts. Further, noting that the signature one day may become a diagnostic tools to identify CDC, is worthwhile.

Author Response

We thank the reviewer for the comments. We have updated the Discussion section with sentences related to future directions regarding in vitro validation of our findings and translation of the INT-CDC signature into a clinical diagnostic tools (pages 17 and 18).

Reviewer 2 Report

I have no further comments and suggestions. It is too bad not to be able show representative histology, but it seems unavoidable in this case.  Very interesting work, thank you.  

Author Response

We thank the reviewer for his valuable comments.